# NADPH Dynamics: Linking Insulin Resistance and β-Cells Ferroptosis in Diabetes Mellitus

**DOI:** 10.3390/ijms25010342

**Published:** 2023-12-26

**Authors:** Dong-Oh Moon

**Affiliations:** Department of Biology Education, Daegu University, 201 Daegudae-ro, Gyeongsan-si 38453, Gyeongsangbuk-do, Republic of Korea; domoon@daegu.ac.kr; Tel.: +82-53-850-6992

**Keywords:** NADPH, insulin resistance, ferroptosis, ROS

## Abstract

This review offers an in-depth exploration of Nicotinamide Adenine Dinucleotide Phosphate (NADPH) in metabolic health. It delves into how NADPH affects insulin secretion, influences insulin resistance, and plays a role in ferroptosis. NADPH, a critical cofactor in cellular antioxidant systems and lipid synthesis, plays a central role in maintaining metabolic homeostasis. In adipocytes and skeletal muscle, NADPH influences the pathophysiology of insulin resistance, a hallmark of metabolic disorders such as type 2 diabetes and obesity. The review explores the mechanisms by which NADPH contributes to or mitigates insulin resistance, including its role in lipid and reactive oxygen species (ROS) metabolism. Parallelly, the paper investigates the dual nature of NADPH in the context of pancreatic β-cell health, particularly in its relation to ferroptosis, an iron-dependent form of programmed cell death. While NADPH’s antioxidative properties are crucial for preventing oxidative damage in β-cells, its involvement in lipid metabolism can potentiate ferroptotic pathways under certain pathological conditions. This complex relationship underscores the delicate balance of NADPH homeostasis in pancreatic health and diabetes pathogenesis. By integrating findings from recent studies, this review aims to illuminate the nuanced roles of NADPH in different tissues and its potential as a therapeutic target. Understanding these dynamics offers vital insights into the development of more effective strategies for managing insulin resistance and preserving pancreatic β-cell function, thereby advancing the treatment of metabolic diseases.

## 1. Introduction

Diabetes mellitus is a widespread metabolic disorder where the body experiences elevated blood glucose levels due to either insufficient insulin production or the body’s impaired ability to use insulin [1,2]. Type 1 diabetes mellitus (T1DM), an autoimmune condition, occurs when the body’s immune system mistakenly attacks and destroys insulin-producing cells in the pancreas, leading to a critical deficit of insulin [3,4]. In Type 2 diabetes mellitus (T2DM), the body develops resistance to insulin, and despite having adequate or increased levels of this hormone, it becomes less effective at managing blood glucose levels [3,4].

Insulin resistance is a multifaceted disorder that is central to the development of T2DM and is closely associated with a cluster of cardiovascular risk factors known as metabolic syndrome [5,6]. At the cellular level, insulin resistance implicates a disruption in insulin signaling pathways, leading to impaired glucose uptake and metabolism [7]. Emerging evidence has pinpointed the role of NADPH, a pivotal cofactor in cellular redox reactions, in the modulation of insulin action [8,9].

NADPH is primarily known for its role in biosynthetic reactions and the maintenance of redox balance by regenerating reduced glutathione [10,11]. However, recent studies have unveiled its significance in the regulation of oxidative stress and inflammatory responses, which are key contributors to the pathogenesis of insulin resistance [12,13]. The oxidative burden in insulin resistance is characterized by an imbalance between pro-oxidants and antioxidant defenses, leading to the oxidative modification of cellular components, which further impairs insulin signaling [14].

Moreover, NADPH oxidases (NOXs), which utilize NADPH as a substrate to produce ROS, have been implicated in the development of insulin resistance. The overactivation of NOX enzymes exacerbates oxidative stress, and in parallel, the consequential depletion of NADPH reserves may impair the cell’s antioxidant capacity, setting a vicious cycle that perpetuates insulin resistance [15]. In adipocytes, this oxidative stress has been linked to the dysregulation of adipokines, which are crucial in modulating insulin sensitivity [16].

Furthermore, the role of NADPH in lipid metabolism, particularly in the fatty acid synthesis pathway, ties it to the etiology of insulin resistance, where lipid accumulation and lipotoxicity are known to disrupt insulin signaling [17]. The interplay between NADPH levels and the endoplasmic reticulum stress response also provides a connection to insulin resistance, as this stress response is activated in conditions of nutrient excess and metabolic disturbance [18].

The current body of research suggests that targeting NADPH-related pathways could offer new therapeutic avenues for the treatment of insulin resistance. By understanding the nuances of NADPH’s role in insulin signaling, the scientific community can move closer to unraveling the complex web of metabolic dysfunction that characterizes insulin resistance.

In this review, we delve into the molecular intricacies of NADPH’s involvement in insulin resistance, exploring the evidence from cellular studies to clinical research that highlights its dual role as both a culprit in pathogenesis and a potential target for therapeutic intervention.

## 2. Biological Role of NADPH

NADPH functions as a coenzyme essential to cellular metabolism, playing pivotal roles in various critical biological processes. NADPH’s primary role is in scavenging ROS. Within cells, superoxide anion (O_2_^−^) is generated and subsequently transformed into H_2_O_2_. Glutathione peroxidase (GPX) and glutathione reductase (GR) then detoxify H_2_O_2_, with NADPH supplying the required reducing equivalents for this antioxidant activity [19]. This process is vital for shielding cells from oxidative harm by neutralizing damaging ROS.

Another function of NADPH is in ROS production. NOXs on the cell membrane generate O_2_^−^ by transferring electrons from NADPH inside the cell to molecular oxygen (O_2_) [20]. This superoxide is then converted into H_2_O_2_, either spontaneously or through enzymatic action by superoxide dismutase (SOD).

NADPH also plays a crucial role in fatty acid synthesis. The process begins with acetyl-CoA carboxylase (ACC) converting acetyl-CoA to malonyl-CoA, a fatty acid precursor. Fatty acid synthase (FAS) then elongates malonyl-CoA into palmitate in a NADPH-dependent manner [21]. Additionally, Stearoyl-CoA desaturase-1 (Scd1) introduces a double bond into saturated fatty acids, transforming them into monounsaturated fatty acids with distinct biological properties. For triglyceride synthesis, glycerol-3-phosphate acyltransferase (GPAT) esterifies glycerol-3-phosphate with a fatty acid to form lysophosphatidic acid. This compound is further modified by 1-acylglycerol-3-phosphate acyltransferase (AGPAT) into phosphatidic acid, which is then dephosphorylated by Phosphatidic acid phosphohydrolase-1 (PPH-1) or lipin 1, to yield diacylglycerol (DAG). Diacylglycerol acyltransferase (DGAT) completes triglyceride synthesis by attaching an additional fatty acid to DAG [22].

Lastly, NADPH is involved in cholesterol synthesis, beginning with the conversion of acetyl-CoA to 3-hydroxy-3-methylglutaryl-CoA (HMG-CoA) by HMG-CoA synthetase. HMG-CoA reductase then reduces HMG-CoA to mevalonate, a key step in cholesterol synthesis, utilizing NADPH [23]. Mevalonate is further processed into isopentenyl pyrophosphate (IPP) and dimethylallyl pyrophosphate (DMAPP), leading to cholesterol formation. Additionally, NADPH is instrumental in synthesizing several non-essential amino acids through reductive amination. A summary of NADPH’s functions is illustrated in Figure 1.

## 3. Biosynthetic Pathway of NADPH

Cellular production of NADPH is facilitated through several pathways, including the pentose phosphate pathway, the citric acid cycle, and fatty acid metabolism. The dynamic equilibrium between NADPH synthesis and consumption is essential for preserving cellular redox balance and enabling a host of biosynthetic reactions.

NADPH is primarily synthesized through the action of NAD kinase (NADK), which phosphorylates NAD^+^ to create NADP^+^. This NADP^+^ is then reduced to NADPH by various dehydrogenases/reductases across different metabolic pathways [24,25]. NADKs are ubiquitously present in almost all human tissues, excluding skeletal muscle, and are found in both the cytosol and mitochondria. The mitochondrial NADK (mNADK) is unique in its ability to also phosphorylate NADH, directly producing NADPH within mitochondria to mitigate oxidative stress [26].

The pentose phosphate pathway branches from glycolysis and is a principal source of cytosolic NADPH through three non-reversible steps in its oxidative phase [27,28]. Enhanced glucose flux into this branch has been linked to increased NADPH production in various cancers [29]. The enzyme glucose-6-phosphate dehydrogenase (G6PD) catalyzes the conversion of glucose-6-phosphate (G6P) to 6-phosphogluconolactone (6-PGL), simultaneously generating NADPH [30]. The subsequent reaction, led by 6-phosphogluconate dehydrogenase (PGD), produces ribulose-5-phosphate (Ru5P) and another molecule of NADPH [31].

The folate-mediated one-carbon metabolism is well-known for producing one-carbon units for nucleic acid and methionine synthesis and also for generating NADPH [32]. Serine and glycine serve as the main carbon donors in this pathway, with activated serine biosynthesis contributing to increased NADPH generation in cells [33]. Enzymes like methylene tetrahydrofolate dehydrogenases (MTHFD1 in the cytosol and MTHFD2/MTHFD2L in mitochondria) and 10-formyl-THF dehydrogenases (ALDH1L1 in cytosol and ALDH1L2 in mitochondria) are involved in reactions that produce NADPH. MTHFD2 is considered a key driver for rapid growth in mitochondria, affecting the response to certain chemotherapeutic agents [34].

Malic enzymes (MEs) are pivotal in linking glycolysis and the Krebs cycle, facilitating anabolic metabolism and NADPH production through the conversion of malate to pyruvate [35]. Quantitative flux analysis suggests that ME’s contribution to NADPH generation is comparable to that of the pentose phosphate pathway [36]. The ME family has three isoforms: ME1 in the cytosol and ME2 and ME3 in mitochondria.

Isocitrate dehydrogenases (IDH) aid in NADPH production from NADP^+^ by converting isocitrate to α-ketoglutarate (α-KG), a key step in the TCA cycle [37]. IDH has three isoforms: IDH1 in the cytosol and peroxisomes, and IDH2/3 in mitochondria, with IDH1/2 using NADP^+^ for reversible reactions and IDH3 using NAD^+^ for an irreversible process [38].

Glutamine plays a multifaceted role in cellular metabolism: it contributes to the TCA cycle as a carbon source, provides nitrogen for the synthesis of nucleotides, amino acids, and lipids, and is essential for maintaining NADPH levels [39]. The process of glutaminolysis starts in the mitochondria, where glutamine is converted to glutamate by glutaminases (GLS1/2). This glutamate is then transformed into α-ketoglutarate (α-KG) by NADPH-dependent glutamate dehydrogenases (GDH) or transaminases such as GOT2 and GPT2, facilitating the production of necessary amino acids [40].

Nicotinamide nucleotide transhydrogenase (NNT) is a key protein in mitochondrial membranes that catalyzes the hydride ion transfer from NADH to NADP^+^, producing NADPH by leveraging the proton gradient created by the electron transport chain (ETC) [41]. It plays a vital role in maintaining mitochondrial NADPH and NADH levels, contributing to about 45% of the mitochondrial NADPH pool [42]. The NADPH biosynthetic pathway is shown in Figure 2.

## 4. Changes in Expression of NADPH Producing Enzymes in Diabetes Mellitus

In diabetes, changes in glucose, amino acids, and fatty acids occur. Elevated glucose levels in diabetes can lead to increased NADPH production through the pentose phosphate pathway [43,44]. However, prolonged hyperglycemia can cause oxidative stress, potentially depleting NADPH needed for antioxidant defense, such as regenerating reduced glutathione [45]. Amino acids, particularly branched-chain amino acids (BCAAs), including leucine, isoleucine, and valine, are known to be increased in diabetes, influencing insulin secretion and metabolic regulation through upregulation of NADPH production [46]. Fatty acids also play a significant role, as they are involved in various metabolic processes related to diabetes, affecting insulin sensitivity and glucose metabolism. 

This review brings together the findings from multiple studies examining the alterations in the expression and activity of NADPH-producing enzymes in the context of diabetes mellitus, a condition marked by metabolic dysregulation. A consistent theme across these studies is the upregulation of key enzymes involved in NADPH production, which plays a crucial role in cellular redox balance and biosynthetic processes.

Seo et al., and Fomina have both highlighted significant increases in the expression and activity of G6PD in streptozotocin-induced diabetic mice [47,48]. G6PD, a pivotal enzyme in the pentose phosphate pathway, showed elevated levels in various tissues, suggesting a systemic response to the hyperglycemic state induced by diabetes. Further emphasizing the impact of diabetes on the pentose phosphate pathway, Ulusu et al., demonstrated increased activity of PGD in a diabetic model [49]. This increase in PGD activity points towards an adaptive response to the heightened oxidative stress commonly associated with diabetes.

The studies by Yang et al., and Li et al., have provided insights into the upregulation of ME1 in the context of diabetes. Elevated ME1 expression was observed in patients with diabetic peripheral neuropathy and T2DM, respectively [50,51]. Additionally, Li et al., reported an increase in ME2 expression in T2D patients [51]. These findings suggest a broader shift in metabolic pathways linking glycolysis and the citric acid cycle in diabetes.

The research by Lee et al., and Wang et al., brings to light the altered expression and activity of IDH1 and IDH2 in diabetic models [44,52]. Elevated levels of IDH1 were noted in diabetic rodents, while increased IDH2 levels were observed in mice on a high-fat diet. These changes in IDH enzymes, crucial for the citric acid cycle and glutaminolysis, indicate a reconfiguration of metabolic pathways in response to diabetic conditions.

Vohra et al., observed a significant 2.9-fold increase in the expression of Methylenetetrahydrofolate Dehydrogenase 1 (MTHFD1) in T2D patients [53]. This increase in MTHFD1, an enzyme involved in folate metabolism, underscores the metabolic alterations accompanying diabetes.

Lastly, King et al., reported a notable 2.0-fold increase in the expression of N-Myristoyltransferase (MNT) in a diabetic model [54]. Although MNT is primarily known for its role in protein modification, its upregulation in diabetes indicates an adaptation of cellular processes to the metabolic challenges posed by this condition.

In summary, these studies collectively underscore a significant upregulation in the enzymes involved in NADPH production in various models of diabetes, reflecting a systemic metabolic response to the challenges posed by this chronic condition. The relationship between diabetes and NADPH biosynthesis enzymes is summarized in Table 1.

## 5. Role of NADPH-Producing Enzymes in Diabetes

In the field of metabolic research, recent studies have shed light on the intricate relationships between various enzymes and insulin regulation, with a focus on G6PD, 6PGD, ME1, ME3, IDH1, IDH2, GDH1, and GDH2.

Jiang and colleagues demonstrated that overexpression of G6PD in C2C12 mouse myoblast cell lines results in insulin resistance [55]. Similar findings were reported by Wang, who linked G6PD overexpression in Wistar rats, induced by a high-fat diet, to enhanced insulin resistance [56]. Park and his team found that G6PD overexpression in adipocytes upregulates insulin resistance [57]. Contrarily, Monte Alegre S and associates observed that patients with G6PD deficiency exhibit decreased insulin secretion [58]. This paradox is further complicated by the fact that G6PD overexpression can also impair glucose-stimulated insulin secretion due to increased NOX expression and ROS accumulation [59]. Moving to 6PGD, research by Spegel et al., indicates that inhibiting 6PGD blocks glucose-stimulated insulin secretion in β-cells, highlighting its role in insulin regulation [60].

In the context of ME1, Al-Dwairi et al., found that ME1-expressing transgenic mice exhibit enhanced insulin resistance, implicating ME1 in glucose metabolism and insulin sensitivity [61]. ME3’s role was explored by Hasan and colleagues, who discovered that ME3 si-RNA treatment decreases insulin release in CHS cell lines, suggesting a regulatory role in insulin secretion [62].

The role of IDH1 in insulin regulation was demonstrated by Guay and colleagues, who showed that IDH1 knockdown reduced insulin release in INS-1 832/13 cell lines [63]. Furthering this, Lee et al., revealed that insulin sensitivity is enhanced in IDH2 knockout mice [64]. Additionally, IDH2 knockdown also decreased insulin release in INS-1 832/13 cell lines [52], indicating its dual role in insulin sensitivity and secretion.

Kim et al., conducted an in-depth investigation revealing that mitochondrial NAD kinase (mNADK) knockout enhances insulin resistance [65]. This study delved into the metabolic mechanisms, demonstrating how the absence of mNADK disrupts the NADP/NADPH balance in cells, thereby influencing insulin signaling pathways. Kim’s research also explored the downstream effects of this knockout on glucose metabolism and energy homeostasis, providing a broader understanding of how mNADK plays a crucial role in maintaining insulin sensitivity and potentially implicating it in the development of metabolic disorders like T2DM.

In the domain of GDH1, research by Vetterli et al., showed that GDH1 knockout leads to decreased obesity in diet-induced mice, suggesting a potential therapeutic target for obesity-related insulin resistance [66]. Lastly, research on GDH2 by Petraki and colleagues revealed that GDH2 knockin mice exhibit enhanced fasting serum insulin levels, pointing to its influence on fasting insulin regulation [67].

Collectively, these studies underscore the complex and multifaceted roles that these enzymes play in insulin regulation, offering potential new avenues for understanding and treating metabolic disorders related to insulin resistance and diabetes. The functions of NADPH biosynthetic enzymes are shown in Table 2.

## 6. The Role of NADPH in Affecting Insulin Secretion

Insulin secretion issues are present in both Type 1 and Type 2 DM due to distinct mechanisms. Type 1 DM is characterized by an autoimmune destruction of pancreatic β-cells, leading to a deficiency in insulin production [68,69]. On the other hand, Type 2 DM is predominantly associated with insulin resistance, a condition where cells progressively lose their responsiveness to insulin. This insulin resistance can eventually strain the pancreatic β-cells, reducing their ability to produce insulin [70,71]. Consequently, both types of diabetes result in ineffective blood glucose regulation due to insufficient insulin activity.

The processes governing insulin secretion from pancreatic β-cells revolve around a dynamic metabolic sequence, where the conversion of glucose into energy alters the ATP/ADP ratio, setting off a series of cellular reactions leading to insulin release. This process begins with the inhibition of ATP-sensitive potassium channels, followed by depolarization that activates voltage-dependent calcium channels [72,73]. Subsequently, the influx of calcium triggers the exocytosis of secretory vesicles that release insulin. Crucially, this pathway is enhanced by metabolic routes, notably the pentose phosphate pathway, in which NADPH plays a pivotal role. 

Recent research exploring the connection between insulin secretion and NADPH in pancreatic β-cells highlights a complex interplay. These studies investigate how NADPH, crucial for cellular redox balance and biosynthetic processes, influences insulin release mechanisms in these cells. Kalwat and Cobb’s exploration into this pathway shed light on β-cell metabolic capacities to enhance insulin release beyond the initial calcium-induced response [74]. This is complemented by Ferdaoussi and colleagues’ demonstration of how isocitrate influences Sentrin-specific protease 1 (SENP1) signaling with NADPH’s assistance, significantly increasing insulin secretion and presenting a promising target for T2DM therapy [75]. The work of Monte Alegre et al., further emphasizes the essential role of G6PD in this pathway, as its deficiency correlates with decreased insulin secretion [58]. Spegel and his team’s association of the pentose phosphate pathway with insulin secretion underscore the importance of the NADPH/NADP^+^ ratio as an indicator of β-cell metabolic response, where any disturbances could impede insulin release [60]. Zhang’s research highlights the protective role of NADPH in β-cell health, as elevated glucose levels impede G6PD activity, leading to oxidative stress and β-cell apoptosis [76]. Lee’s findings reveal that increased G6PD activity might exacerbate ROS buildup, contributing to β-cell impairment in T2DM [59]. Riganti et al.’s insights into the antioxidant defense of the pentose phosphate pathway offer valuable perspectives on β-cell redox stability [77]. Plecitá-Hlavatá and her team underscored the importance of NOX4 in generating H_2_O_2_, a vital signal for insulin secretion, indicating that redox signaling is a fundamental part of the secretion process [78]. However, there are also conflicting findings regarding the role of NOXs. Ning Li et al.’s study demonstrates that NOX2, an isoform of NADPH oxidase present in human and mouse β-cells, acts as a negative modulator of insulin secretion. Nox2 deficiency in pancreatic islets leads to enhanced insulin release when stimulated with high glucose levels. This effect is linked to lower superoxide levels and increased cAMP concentrations. The study suggests that NOX2 in β-cells reduces cAMP/PKA signaling through ROS generation, with a noted reciprocal inhibition between the cAMP/PKA pathway and ROS. This identifies a novel role for NOX2 in regulating the secretory response of pancreatic β-cells [79]. Finally, Santos et al., introduced a paradigm shift with their findings on the reverse operation of NNT, significantly impacting insulin secretion by regulating mitochondrial NADPH and glutathione redox states in response to glucose [80]. The mechanism by which NADPH affects insulin secretion is illustrated in Figure 3. 

Collectively, these studies present a multifaceted view of NADPH’s role within β-cells, not just as a facilitator of calcium signaling but as a central figure in a network of redox reactions and metabolic signaling governing insulin secretion. Effective therapeutic strategies for enhancing β-cell function or managing diabetes must therefore consider the critical and complex role of NADPH and its related pathways, where it serves not merely as a metabolic coupling factor but as a custodian of β-cell integrity, delicately managing the interplay between metabolic inputs and insulin output.

## 7. Insulin Resistance and NADPH

The insulin signaling pathways begin when insulin binds to the α subunit of the insulin receptor (IR), altering the β subunit’s shape. This subunit, which has tyrosine kinase activity, then undergoes autophosphorylation, boosting its kinase function [81,82]. The active IR phosphorylates insulin receptor substrates (IRS), activating phosphatidylinositol 3-kinase (PI_3_K). PI_3_K, in turn, phosphorylates phosphatidylinositol 4,5-bisphosphate (PIP_2_) to form phosphatidylinositol 3,4,5-triphosphate (PIP_3_). This PI_3_K activation triggers a cascade involving various serine kinases like phosphoinositide-dependent kinase-1 (PDK1) and protein kinase B (Akt), leading to the diverse biological effects and metabolic actions of insulin, including the promotion of glucose uptake through the translocation of GLUT4. Additionally, the SH2 domain of Growth factor receptor-bound protein 2 (Grb-2) binds to IR, which activates Sos and, subsequently, the Ras protein. Ras activation initiates a signaling cascade that stimulates Raf, MEK, and MAP kinase (MAPK), which oversees cellular growth, mitogenesis, and differentiation. Insulin resistance refers to a condition in which insulin signaling is blocked, and cells do not respond to the normal action of insulin. 

Persistent inflammation resulting from chronic obesity is a significant contributor to insulin resistance in skeletal adipocytes, muscles, and liver, linked to obesity [83,84]. In this context, M1 macrophages in proximity to adipocytes secrete pro-inflammatory mediators like tumor necrosis factor-α (TNF-α) and interleukin 1β (IL-1β). These cytokines trigger signaling through the c-Jun N-terminal kinase (JNK) and NF-kB pathways. Subsequently, the activation of NF-kB leads to an upregulation of NOX expression. It has been suggested that one of the major sources of cellular ROS in obese adipose tissue is pro-oxidative enzymes, NOX [85,86]. The NOX enzyme family, comprising NOX 1–5, specializes in generating O_2_^−^ or H_2_O_2_ [85,86]. The activation of NOXs 1–3 involves the assembly of both membrane-bound and cytosolic subunits, which are similar or related to the classic phagocytic subunits like p22phox, p47phox, and p67phox, leading to O_2_^−^ production [87,88]. Unlike these, Nox4 pairs with p22phox and relies on Poldip2 in the cytoplasm for its function, with evidence of its ability to generate both O_2_^−^ and H_2_O_2_. Nox5’s activity is regulated by calcium binding to its EF hand motifs and does not depend on cytosolic subunits, capable of producing either or both O_2_^−^ and H_2_O_2_. NOX enzymes are ubiquitous in cells and play crucial roles in both healthy physiological processes and disease signaling. Skeletal muscle cells express a variety of NOXs, including Nox1, Nox2, and Nox4, while adipocytes primarily express NOX4 and NOX2. In the context of insulin resistance, NOX-derived ROS activate several serine/threonine kinases, including JNK, IκB kinase (IKK), p38 MAP kinase, and protein kinase C (PKC) [89]. Activated JNK and IKK can inhibit insulin signaling through serine phosphorylation of IRS1, which disrupts insulin receptor-mediated tyrosine phosphorylation of IRS1, essential for the propagation of insulin’s metabolic actions. P38 MAP kinase and PKC similarly phosphorylate IRS proteins, attenuating their activity and contributing to negative regulation of insulin signaling. This aberrant phosphorylation pattern of IRS proteins by kinases activated by NOX-derived ROS is a recognized mechanism underlying the development of insulin resistance. 

An alternative pathway leading to insulin resistance involves the activation of lipid metabolism processes. Elevated levels of DAG activate specific isoforms of PKC, such as PKCθ and PKCε. This activation can also stimulate c-Jun N-terminal kinase (JNK) and IkB kinase-β (IKK-β). These kinases preferentially phosphorylate the serine residues on IRS1, which leads to a decrease in their tyrosine phosphorylation. The altered phosphorylation status of IRS1 diminishes the activity of the PI_3_K and Akt pathway [90,91]. As Akt activity decreases, the translocation of glucose transporter type 4 (GLUT4) to the membrane is hindered, resulting in a reduced uptake of glucose into the cells. phosphatidic acid (PA) is another lipid molecule that is used for triglyceride synthesis. Elevated levels of PA have been associated with the inhibition of the mammalian target of rapamycin (mTOR), a central regulator of cell growth and metabolism. The down-regulation of mTOR by PA can contribute to inhibiting Akt phosphorylation and the dysregulation of insulin signaling, adding another layer of complexity to the development of insulin resistance. Moreover, ceramide, a sphingolipid formed from fatty acid metabolism, can negatively regulate Akt, a serine/threonine-specific protein kinase that is crucial for glucose uptake in response to insulin [92]. The inhibition of Akt by ceramide impairs the insulin signaling pathway, further contributing to insulin resistance. These interactions underline a sophisticated and highly regulated system where lipid metabolism can influence insulin action. The modulation of insulin signaling by lipid metabolites such as PA, DAG, and ceramide demonstrates the intricate balance between lipid homeostasis and glucose metabolism, revealing potential metabolic points of intervention for improving insulin sensitivity and combating metabolic disorders.

Enzymes that produce NADPH, including G6PD, ME, and IDH, are crucial in maintaining redox balance and supporting anabolic activities like de novo lipid synthesis. These enzymes, notably G6PD, ME, and IDH, are found in high concentrations in adipose tissue [93]. While there is an elevated expression of these enzymes in adipocytes of diabetic individuals, the specifics of the signaling pathways involved remain unclear, though they are likely influenced by proinflammatory cytokines. NADPH generated by these enzymes is implicated in ROS production via NOX and is also utilized in the synthesis of new fatty acids, which contributes to the development of insulin resistance.

Furthermore, pro-inflammatory cytokines stimulate lipolysis in adipocytes, resulting in increased circulating levels of free fatty acids (FFAs). These FFAs diminish insulin-stimulated glucose uptake in skeletal muscles, thereby contributing to skeletal muscle insulin resistance [57]. Conversely, lipolysis in adipose tissue boosts hepatic acetyl CoA concentrations and activates pyruvate carboxylase, enhancing hepatic glucose production. This increase in hepatic glucose output, along with pro-inflammatory cytokines, plays a role in the development of hepatic insulin resistance [94,95]. In the liver, FFAs also escalate ROS production, leading to exacerbated insulin resistance and the potential development of nonalcoholic steatohepatitis (NASH) [96,97]. Additionally, prolonged exposure to high levels of FFAs can lead to dysfunction and eventual death of pancreatic β-cells [98,99]. The effect of NADPH on insulin resistance is presented in Figure 4.

## 8. Ferroptosis and NADPH in T2DM

Elevated blood sugar and pro-inflammatory cytokines activate various signaling pathways, culminating in the death of pancreatic β-cells through mechanisms such as necroptosis, apoptosis, ferroptosis, and necrosis [100]. This review focuses on understanding ferroptotic cell death in relation to NADPH in T2DM.

Ferroptosis is a unique form of cell death, distinguished by the accumulation of lipid peroxides and dependent on iron [101]. Iron, an essential element, is typically transported in the blood bound to transferrin. Cells uptake this iron-transferrin complex via the transferrin receptor, a process that is tightly regulated. Once internalized, iron is released from transferrin in an acidic endosomal environment and then exported into the cytoplasm, where it undergoes oxidation. This oxidation state of iron is crucial for its involvement in various biochemical processes, including the Fenton reaction. In the Fenton reaction, ferrous iron (Fe^2+^) reacts with H_2_O_2_, producing highly reactive hydroxyl radicals [102]. These radicals can cause significant cellular damage, contributing to oxidative stress. Lipoxygenases are enzymes that insert molecular oxygen into polyunsaturated fatty acids, creating lipid hydroperoxides [103]. These enzymes require iron for their catalytic activity. Similarly, cytochrome P450 oxidoreductase also utilizes iron in its active site to facilitate the oxidation of organic substrates, including drugs and steroids.

The signal transduction of pro-inflammatory cytokines, specifically TNF-α and IL-1β, plays a pivotal role in the expression of NOX and the generation of ROS. TNF-α and IL-1β are key mediators of inflammation and can activate various cellular signaling pathways, leading to the induction of NOX expression. NOX enzymes are responsible for the production of ROS, which are reactive molecules that can modify cellular components like lipids, proteins, and DNA. The ROS generated by NOX, particularly O_2_^−^, can further interact with iron molecules within the cell. This interaction can lead to the oxidation of iron, a process that is exacerbated in conditions of iron overload or dysregulated iron metabolism. The oxidation of iron can amplify the production of harmful ROS through reactions like the Fenton reaction, where Fe^2+^ reacts with hydrogen peroxide to produce hydroxyl radicals, one of the most reactive forms of ROS. 

Ferroptosis is also closely linked to the peroxidation of polyunsaturated fatty acids (PUFAs) and subsequent lipid peroxidation [104]. This process begins when PUFAs incorporated in cellular membranes are oxidized, forming PUFA hydroperoxides (PUFA-OOH). This oxidation is typically facilitated by iron, which acts as a catalyst. Lipid peroxidation involves a chain reaction of oxidative degradation of lipids. Once initiated, it leads to the production of a variety of reactive aldehydes, most notably malondialdehyde and 4-hydroxynonenal, which can further damage cellular components. This cascade of lipid peroxidation is a hallmark of ferroptosis. It disrupts the integrity of cellular membranes, leading to cell death. Unlike apoptosis, ferroptosis is not marked by caspase activation or DNA fragmentation but is characterized by the accumulation of lipid peroxides and mitochondrial damage.

The defense against ferroptosis involves several key components, namely the glutamate-cystine antiporter (system Xc-), GSH, and GPX4 [105]. System Xc- plays a crucial role in ferroptosis defense by exchanging extracellular cystine for intracellular glutamate. Cystine, once inside the cell, is reduced to cysteine, which is a vital precursor for the synthesis of GSH. GSH is a major cellular antioxidant that helps neutralize ROS and protect cells from oxidative damage. GPX4, a selenoenzyme, utilizes GSH to reduce lipid hydroperoxides to their corresponding alcohols or free hydrogen peroxide to water, thereby preventing the lethal accumulation of lipid peroxides, a key feature of ferroptosis. This mechanism highlights the intricate cellular defense strategies against oxidative stress and iron-mediated toxicity.

NADPH serves a dual role in cellular processes related to ferroptosis. On one hand, it acts as a substrate for NOX, contributing to the generation of ROS, which can promote ferroptosis. On the other hand, NADPH plays a vital role in the regeneration of GSH, an important cellular antioxidant. Through its involvement in the reduction of oxidized glutathione, NADPH supports the activity of GPX4. GR catalyzes the reduction of GSSG to reduced GSH using NADPH as an electron donor, thereby maintaining cellular antioxidant defenses [105]. This action of NADPH helps in suppressing lipid peroxidation and, consequently, ferroptosis, thereby highlighting its complex and dualistic nature in cellular redox balance.

The dual role of NADPH in ferroptosis, influenced by its interactions with NOX and GR, is shaped by various factors, including enzyme affinity, expression, and activity. While specific studies measuring the enzyme affinity of NOX and GR for NADPH in pancreatic β-cells are lacking, existing data provides insights. For example, the Km value of NOX in neutrophils for NADPH is relatively low at 4–7 microM, indicating a high affinity [106]. In contrast, bovine liver GR exhibits a higher Km value of 63 ± 8 microM, suggesting a lower affinity for NADPH compared to NOX [107]. This differential affinity indicates that NOX might bind NADPH more readily than GR. Additionally, the expression of NOX is known to be upregulated by pro-inflammatory cytokines in diabetic conditions, leading to increased ROS production and potentially promoting ferroptosis. On the other hand, while the expression level of GR does not significantly change in diabetes, its enzymatic activity is reported to decrease, which could reduce its effectiveness in combating oxidative stress [108]. These observations suggest that in certain pathological states, like diabetes, NADPH may predominantly fuel processes leading to ferroptosis, mainly due to the enhanced activity and affinity of NOX compared to GR. This hypothesis needs further investigation to be confirmed, as more specific and targeted research in this area is essential to fully understand the dynamics of NADPH in ferroptosis, particularly in the context of diabetes and pancreatic β-cell health. 

Ferroptosis in β-cells results in reduced insulin secretion and β-cell mass, exacerbating the hyperglycemic state. This process is considered a key pathophysiological mechanism in diabetes development and progression. By understanding the mechanisms underlying β-cell ferroptosis, novel therapeutic strategies can be developed to protect β-cells from ferroptosis and potentially slow down or prevent the progression of diabetes. This could include approaches to reduce oxidative stress, iron chelation, or the use of inhibitors targeting the ferroptosis pathway. The impact of NADPH on ferroptosis is illustrated in Figure 5.

## 9. The Role of NOXs in Insulin Secretion, Insulin Resistance, and Ferroptosis

In this review, we delve into the significant roles of NOXs in insulin secretion, insulin resistance, and ferroptosis. These enzymes, which utilize NADPH to generate ROS, are increasingly recognized for their complex impact on cellular functions. We will summarize recent research findings, highlighting how NOXs contribute to these key metabolic processes and their implications in the pathophysiology of metabolic disorders.

In the realm of diabetes research, the effects of NOXs on insulin secretion present a complex picture. While some studies indicate NOXs inhibit insulin secretion [79], others suggest they promote it [78,109,110,111]. This disparity can be attributed to the diverse roles of NOX isoforms and the context-specific nature of their action in pancreatic beta cells. For instance, NOX2 has been shown to negatively modulate insulin secretion, whereas NOX4 is essential for glucose-stimulated insulin secretion. The dualistic nature of NOXs, both as sources of ROS and as modulators of cellular signaling pathways, offers an explanation for these seemingly contradictory findings. Further research is necessary to fully understand the intricate relationship between NOXs and insulin secretion and their potential therapeutic implications in diabetes.

Moving forward in our review, we will explore the research surrounding the influence of NOXs on insulin resistance. This segment focuses on how NOXs, known for their role in generating reactive oxygen species, potentially contribute to the development and progression of insulin resistance in various physiological contexts. Recent research by Souto Padron de Figueiredo and colleagues demonstrated a decrease in insulin resistance in mice with a NOX2 gene knockout [112]. In a similar vein, studies by Den Hartigh et al., revealed a reduction in insulin resistance in mice lacking the NOX4 gene [113]. Furthermore, the use of NOX inhibitors has been shown to lower ROS production and normalize adipokine levels, thus helping to relieve metabolic disorders like insulin resistance in cases of obesity [114,115]. Park et al.’s study also demonstrated that administering APX-115, a pan-Nox inhibitor, in aging diabetic mice significantly improved insulin resistance, reduced oxidative stress indicators like urinary 8-isoprostane, and ameliorated kidney symptoms [116]. This compiled research indicates that NOX enzymes significantly influence insulin resistance. Their modulation, particularly through inhibition, shows potential in managing insulin resistance, a key factor in metabolic disorders like T2DM. Further investigation is needed to fully understand and harness their therapeutic potential.

Lastly, in the context of T2DM and NOXs’ role in ferroptosis, Yao et al., demonstrated that treating high-fat diet-fed mice with Vas2870, a NOX inhibitor, effectively reduced ferroptosis accumulation and altered expression of related proteins such as ASCL4, FTH1, and GPX4 [117]. Wang et al., observed that applying Nox2-siRNA led to reduced ferroptosis [118]. Meanwhile, Weaver et al., reported that treatment with apocynin and DPI, both. 

NOX inhibitors resulted in decreased ROS production and cell death [119]. These findings collectively underscore the potential therapeutic impact of NOX inhibition in managing ferroptosis in T2DM. Table 3 presents an extensive summary of research findings on the role of NOXs in insulin secretion, insulin resistance, and ferroptosis.

## 10. Conclusions

This comprehensive review underscores the critical role of NADPH in the complex landscape of metabolic disorders, particularly insulin resistance and ferroptosis. Our exploration reveals that NADPH, through its involvement in redox reactions and biosynthetic processes, stands at the crossroads of metabolic health and disease. In the context of insulin resistance, NADPH’s role is multifaceted, influencing oxidative stress, inflammatory responses, and lipid metabolism in adipocytes and skeletal muscle. The balance between its antioxidative functions and participation in pathways that exacerbate metabolic dysfunction is a delicate one, underscoring the importance of maintaining cellular redox homeostasis.

The discussion on ferroptosis in pancreatic β-cells further illuminates the dual nature of NADPH. While it is essential for the antioxidant defense mechanisms that protect β-cells, its involvement in pathways leading to iron-dependent lipid peroxidation highlights a paradoxical aspect that could contribute to cell death in diabetic conditions. This intricate relationship between NADPH and ferroptosis emphasizes the need for targeted therapeutic strategies that can modulate NADPH dynamics to protect β-cells.

NADPH also plays a significant role in the development of atherosclerosis and other cardiovascular diseases, as well as in renal disease. It’s involved in oxidative stress processes, which are key factors in these conditions. NADPH contributes to the production of ROS, and imbalances in ROS can lead to damage to blood vessels and kidneys. This can accelerate the progression of diseases like atherosclerosis, which involves the hardening and narrowing of arteries and can impact kidney function. Managing NADPH levels and oxidative stress is therefore important in these diseases.

The findings compiled in this review not only deepen our understanding of NADPH’s role in metabolic health but also open avenues for future research. There is a clear indication that manipulating NADPH-related pathways could offer promising therapeutic approaches for managing insulin resistance and safeguarding pancreatic β-cells. As our understanding of these complex metabolic interactions evolves, we move closer to developing more effective interventions for metabolic diseases such as diabetes, which remain a significant challenge in modern healthcare.

## Figures and Tables

**Figure 1 ijms-25-00342-f001:**
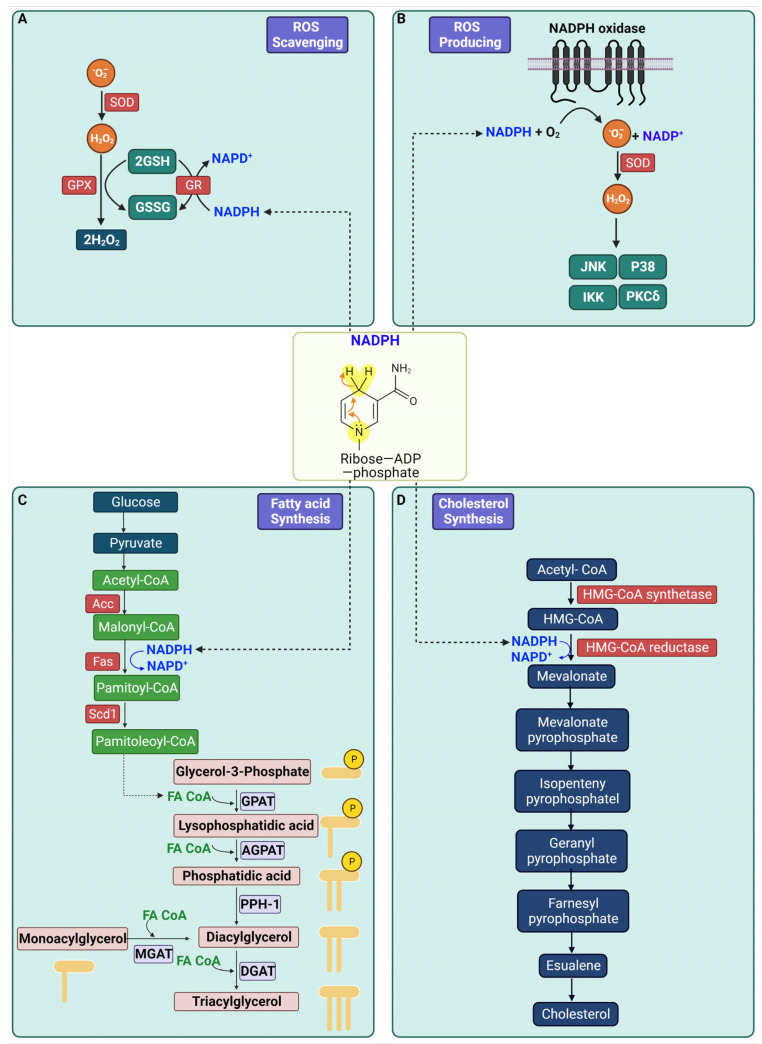
Depiction of NADPH’s Roles in Cellular Metabolism. NADPH is shown as a key coenzyme in cellular processes, primarily in ROS scavenging by transforming and detoxifying O_2_^−^ and H_2_O_2_ and in ROS production via NADPH oxidases. It also plays a crucial role in biosynthesis, aiding in fatty acid elongation and cholesterol synthesis. (**A**) depicts the ROS scavenging mechanisms where NADPH is utilized in the enzymatic reaction that regenerates GSH from its GSSG via GR. This reduced GSH then aids GPX in detoxifying H_2_O_2_ into water, a vital process for controlling oxidative stress within the cell. (**B**) contrasts with the antioxidative role by illustrating NADPH as a substrate for the NADPH oxidase complex, which catalyzes the production of O_2_^−^. These radicals are key signaling molecules that activate various kinases such as JNK, P38, IKK, and PKCδ, thereby implicating NADPH in the modulation of cellular stress responses. (**C**) transitions into the realm of lipid metabolism, detailing NADPH’s essential function in the synthesis of fatty acids. It traces the metabolic flow from glucose to pyruvate, then to acetyl-CoA, and its subsequent carboxylation to malonyl-CoA, followed by elongation to palmitoyl-CoA via fatty acid synthase (Fas), a process reliant on NADPH. The series of reactions leading to the formation of triacylglycerol accentuates the role of NADPH as a reducing agent in the biosynthesis of complex lipids. P represents a phosphate group. (**D**) outlines the cholesterol synthesis pathway, where NADPH again plays a crucial role as a reducing agent, particularly in the conversion of HMG-CoA to mevalonate by HMG-CoA reductase. The pathway is further detailed through subsequent conversions to intermediate compounds like isopentenyl pyrophosphate and farnesyl pyrophosphate, culminating in the production of cholesterol, underscoring the significance of NADPH in the synthesis of this vital lipid molecule. Cartoon in Figure 1 was created with BioRender.com.

**Figure 2 ijms-25-00342-f002:**
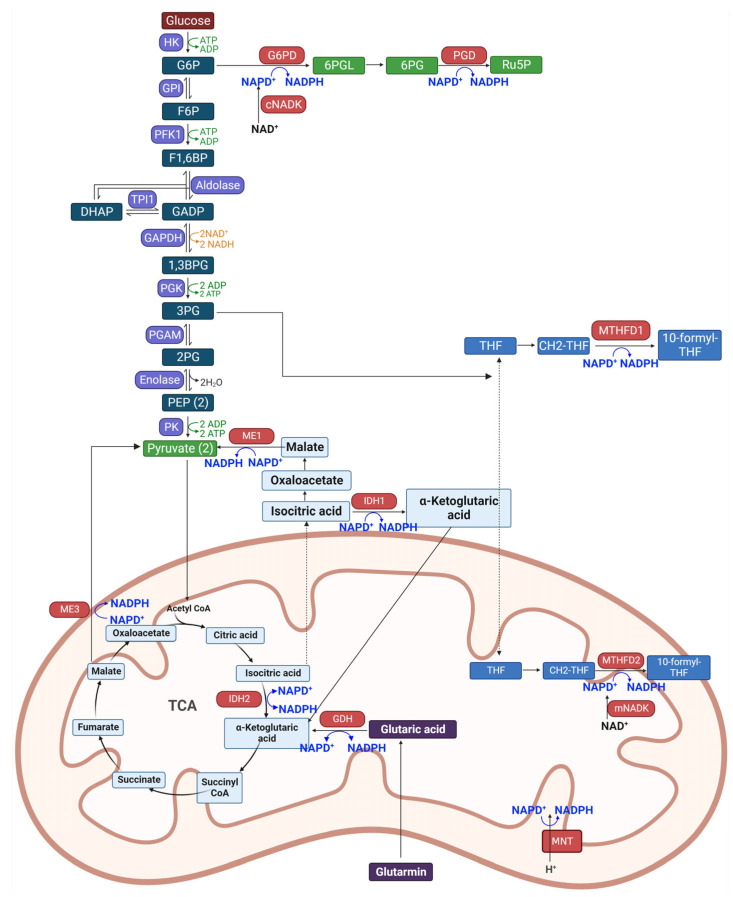
Overview of NADPH Production Pathways in Cells. This figure illustrates the various biochemical pathways involved in the synthesis of NADPH, a crucial coenzyme in cellular metabolism. Key pathways include the pentose phosphate pathway (PPP), the citric acid cycle, and fatty acid metabolism. The PPP, branching from glycolysis, is a major source of cytosolic NADPH, primarily through the action of enzymes like glucose-6-phosphate dehydrogenase (G6PD) and 6-phosphogluconate dehydrogenase (PGD). NADPH is also generated in the folate-mediated one-carbon metabolism, involving enzymes like methylene tetrahydrofolate dehydrogenases (MTHFD1, MTHFD2) and 10-formyl-THF dehydrogenases. Malic enzymes (MEs) and isocitrate dehydrogenases (IDH) further contribute to NADPH production, linking glycolysis with the Krebs cycle. Additionally, glutamine metabolism and nicotinamide nucleotide transhydrogenase (NNT) in mitochondrial membranes play pivotal roles in maintaining NADPH levels. This figure encapsulates the dynamic equilibrium between NADPH synthesis and consumption, essential for cellular redox balance and biosynthesis. Cartoon in Figure 2 was created with BioRender.com.

**Figure 3 ijms-25-00342-f003:**
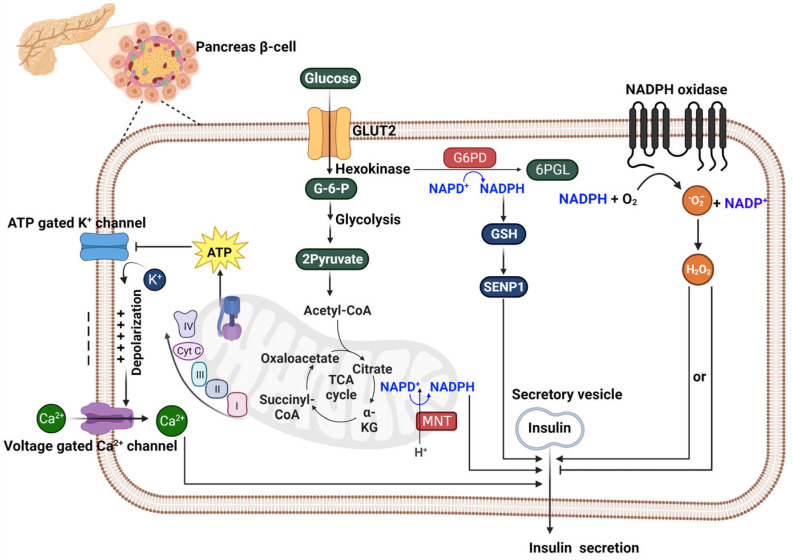
The Role of NADPH in Insulin Secretion from Pancreatic β-Cells. This diagram provides a detailed representation of the biochemical pathways involved in insulin secretion within a pancreatic β-cell. Glucose uptake through the Glucose transporter 2 (GLUT2) transporter initiates a series of metabolic reactions, starting with glycolysis and leading to the production of ATP, which causes the closure of ATP-sensitive potassium channels and subsequent cell depolarization. This depolarization triggers the opening of voltage-gated calcium channels, allowing calcium ions to enter the cell and stimulate the exocytosis of insulin from secretory vesicles. The figure also illustrates the critical role of NADPH in this process: it is generated by the G6PD reaction in the pentose phosphate pathway and is utilized for the regeneration of GSH, with GSH facilitating insulin granule exocytosis via SENP1. Additionally, NADPH produced by NNT in mitochondria also promotes insulin release. However, NADPH is consumed by NADPH oxidase to ROS, which can inhibit the insulin secretion process. The − and + symbols represent membrane potential. Cartoon in Figure 3 was created with BioRender.com.

**Figure 4 ijms-25-00342-f004:**
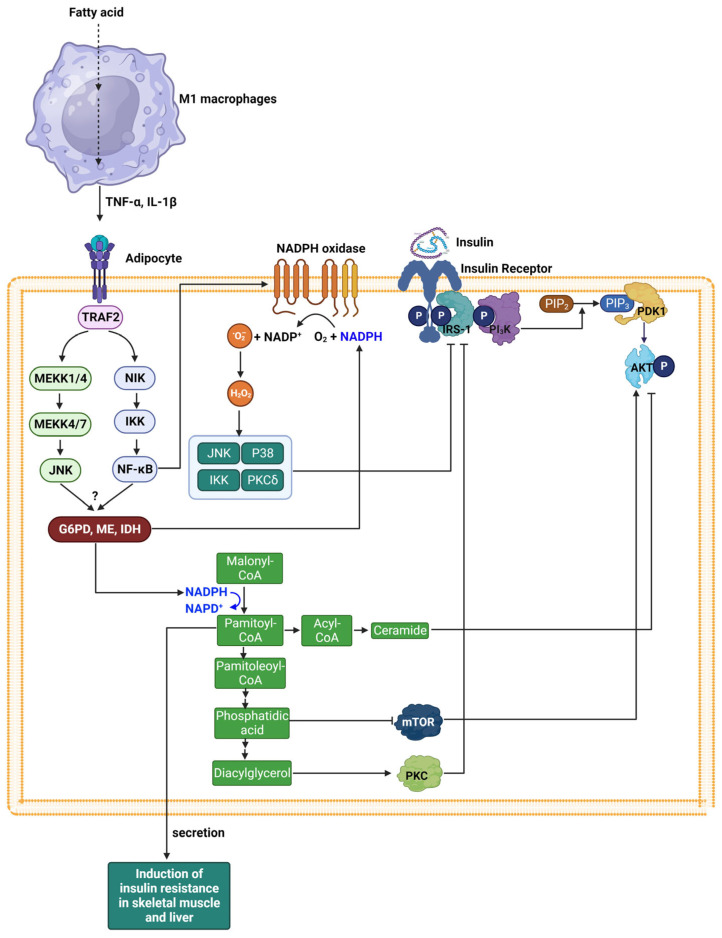
Illustration of Insulin Signaling and Resistance Pathways. This figure depicts the mechanism of insulin signaling, beginning with insulin binding to its receptor, leading to a cascade involving insulin receptor substrates (IRS), PI_3_K, and subsequent activation of various kinases. It also highlights the role of NOX in producing ROS that contribute to insulin resistance, particularly in the context of obesity-induced chronic inflammation. The figure further illustrates how NOX-derived ROS and lipid metabolites like diacylglycerol (DAG) and ceramide disrupt insulin signaling by altering IRS phosphorylation and Akt activity. Additionally, it shows the influence of pro-inflammatory cytokines and free fatty acids (FFAs) on insulin resistance in various tissues, emphasizing the complex interplay between metabolic processes and signaling pathways in insulin resistance. P represents a phosphate group. Cartoon in Figure 4 was created with BioRender.com.

**Figure 5 ijms-25-00342-f005:**
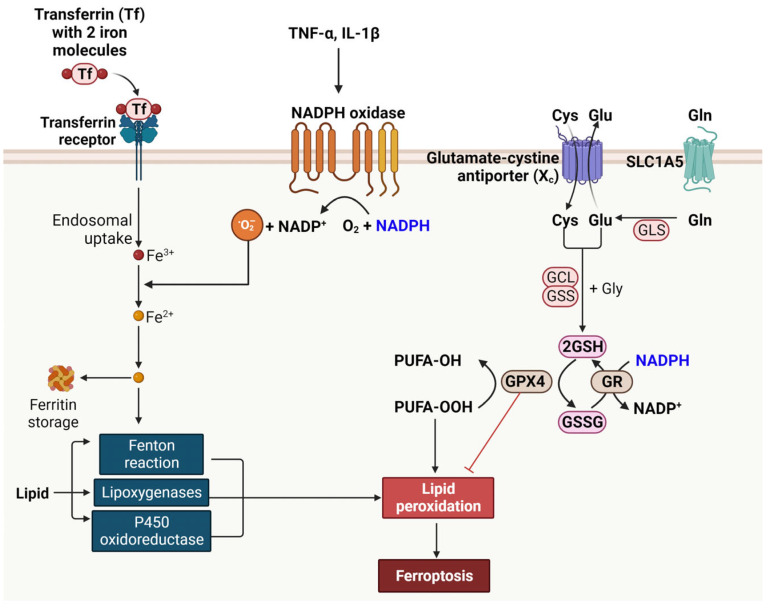
Illustration of Ferroptosis in Pancreatic Beta Cells and Its Relation to NADPH. This figure depicts the process of ferroptosis, a unique form of cell death characterized by iron-dependent lipid peroxidation, in the context of pancreatic beta cells in diabetes. Highlighted are the key elements leading to ferroptosis: elevated blood sugar and pro-inflammatory cytokines triggering oxidative stress and iron accumulation, promoting lipid peroxidation. The figure outlines the role of NADPH in both promoting ROS generation via NOX and supporting antioxidant defense through glutathione regeneration. It showcases the dual function of NADPH in influencing ferroptosis—as a substrate for ROS production and as a crucial element for cellular antioxidant mechanisms. Additionally, the figure underscores the impact of ferroptosis on pancreatic beta cells, including reduced insulin secretion and beta cell mass, contributing to the progression of diabetes. This visualization encapsulates the complex interplay between metabolic factors, oxidative stress, and cell death pathways, emphasizing the importance of understanding ferroptosis in diabetes for developing potential therapeutic interventions. Cartoon in Figure 5 was created with BioRender.com.

**Table 1 ijms-25-00342-t001:** NADPH biosynthesis enzymes in diabetes mellitus. NT = not tested.

Enzyme	mRNA	Protein	Activity	Model	Ref.
G6PD	NT	Elevated G6PD expression in kidney (2.0-fold), lung (1.2-fold), spleen (1.2-fold), and thyroid (3.2-fold)	Elevated G6PD activities in kidney (10.1-fold), liver (1.5-fold), spleen (2.1-fold), and thyroid (3.3-fold)	Streptozotocin-induced diabetic mice	[47,48]
	NT	NT	Elevated activity in kidney (2.0-fold)		
PGD	NT	NT	Elevated enzyme activity	Streptozotocin-induced diabetic mice	[49]
ME1	Elevated expression	NT	NT	Diabetic peripheral neuropathy patients	[50,51]
	NT	Elevated expression	NT	db/db mice	
ME2	NT	Elevated expression	NT	db/db mice	[51]
IDH1	NT	Elevated expression	Elevated activity	Diabetic rats and mice	[44,52]
IDH2	NT	Elevated expression	NT	High-fat diet mice	[44,52]
MTHFD1	Elevated expression (2.9-fold)	NT	NT	T2D patients	[53]
MNT	NT	NT	Elevated activity (2.0-fold)	Streptozotocin-induced diabetic mice	[54]

**Table 2 ijms-25-00342-t002:** Function of NADPH synthesizing enzymes. ↑ and ↓ mean up-regulation and down-regulation, respectively.

Enzymes	Model	Action	Result	Ref.
G6PD	C2C12 mouse myoblast cell line	Over-Expression	Insulin resistance ↑	[55]
	Wistar rats	Over-Expression by high fat diet	Insulin resistance ↑	[56]
	Adipocytes	Over-Expression	Insulin resistance ↑	[57]
	Patients with G6PD deficiency	Over-Expression	Insulin secretion ↓	[58]
	Pancreatic β-cells.	Over-Expression	Insulin secretion ↓	[59]
6PGD	β cells	Glucose stimulation	Insulin secretion ↓	[60]
ME1	Transgenic(Tg) mice	Over-Expression	Insulin resistance ↑	[61]
ME3	CHS cell line	Knock down (si-RNA)	Insulin release ↓	[62]
IDH1	INS-1 832/13 cell line	Knockdown	Insulin release ↓	[63]
IDH2	Mice	Knockout (IDH2^−/−^)	Insulin resistance ↓	[64]
			(Insulin sensitivity ↑)	
	INS-1 832/13 cell line	IDH2 knockdown	Insulin release ↓	[52]
NAD kinase	Mice	Knockout	Insulin resistance ↑	[65]
GDH1	Mice	Knockout	Diet-induced Obesity ↓	[66]
GDH2	Mice	Knockin	Fasting serum insulin levels ↑	[67]

**Table 3 ijms-25-00342-t003:** Effect of NOX inhibition on insulin secretion, insulin resistance, and ferroptosis.

Model	Method	Result	Ref.
Rat Pancreatic Islets, RINm5F cells	Treatment with DPI (NOX inhibitor)	Suppressed insulin secretion in response to high glucose	[109]
Rat Pancreatic Islets	Treatment with DPI; Antisense Oligonucleotide for p47(PHOX)	Decreased ROS production, inhibited glucose-stimulated insulin secretion, reduced glucose oxidation	[110]
Pancreatic Islets, INS-1E Cells	Knockout (NOX4βKO), Silencing (NOX4)	Suppressed insulin secretion in response to high glucose	[78]
Mouse and Human Pancreatic Islets	Nox2 Isoform-Specific Knockout; In Vitro Knockdown of Nox2	Potentiation of insulin release in Nox2-deficient islets compared with controls	[79]
Human Islets	NOX5 knockdown	Suppressed insulin secretion in response to high glucose	[111]
Nox2-null Mice; C2C12 Myotubes	Nox2 Knockout; shRNA Down-regulation of Nox2 in Cells	Insulin resistance induced by high-fat diet was reduced in Nox2-null mice compared to wild-type. Nox2 knockdown in C2C12 cells prevented insulin resistance induced by high glucose or palmitate.	[112]
C57BL/6 Mice with Specific Adipocyte NOX4 Deletion	NOX4 Knockout in Adipocytes	Delayed insulin resistance	[113]
Obese Mice, Human Subjects, Cultured Adipocytes	Treatment with DPI	Improved insulin resistance and reduced oxidative stress	[114,115]
Aging Diabetic Mice	Treatment with APX-115 (pan-Nox inhibitor)	Improved insulin resistance and reduced oxidative stress	[116]
High-fat diet-fed Mice	Treatment with Vas2870 (NOX inhibitor)	Reduced ferroptosis accumulation and related protein expression (ASCL4, FTH1, and GPX4)	[117]
Rats	Nox2-siRNA	Reduced ferroptosis	[118]
Human Donor Islets, Mouse Islets, Beta Cell Lines	Treatment with apocynin and DPI (NOX inhibitor)	Reduced ROS production and cell death	[119]

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
