# Peer review of "NADPH Dynamics: Linking Insulin Resistance and β-Cells Ferroptosis in Diabetes Mellitus"

_ijms, 2023, doi:10.3390/ijms25010342_

Round 1
Reviewer 1 Report
Comments and Suggestions for Authors
The manuscript describing the role of NADPH in insulin resistance and T2DM has been written comprehensively, however, it lacks a discussion on a few aspects.
1. The role of NADPH in affecting insulin secretion is missing.
2. What is the effect of hyperglycemia and other amino acids like leucine on NADPH?
3. The translational aspect of targeting NADPH in T2DM is missing. The effect of suppressing Nox2/NADPH on pancreas or the effect of knocking out Nox2 on insulin secretion and glucose levels should be discussed in a separate section.
4. Islet cell transplantation has been suggested as a therapy to increase insulin production, what are the author's thoughts on this from the perspective of NADPH?
5. The role of Nox inhibitors should be discussed.
6. There is no distinction between type I and type II DM. Like destruction of islet cells may be due to inflammation, a part has been discussed, but please relate it to NADPH. Further, inflammation also plays a role in TIIDM mainly in the presence of adipocyte hypertrophy-mediated inflammation. Thus, it is important to have two subsections, one for type I and one for type II.
7. Including the translation aspect by discussing the role of and probability of targeting NADPH in atherosclerosis and other cardiovascular diseases and in renal disease will increase the readership and will make the review more interesting.
Author Response
Reviewer 1
The manuscript describing the role of NADPH in insulin resistance and T2DM has been written comprehensively, however, it lacks a discussion on a few aspects.
⟶ Thanks a lot for your detailed review. I tried my best to make changes in the short time I had. Just a heads up, I'll be on a business trip from December 22nd for three days where I can't use any electronics due to security reasons. If there's a need for more revisions, please keep my travel dates in mind.
- The role of NADPH in affecting insulin secretion is missing.
⟶ Section 5 and Figure 3 have been newly added to elucidate the effect of NADPH on insulin secretion.
- What is the effect of hyperglycemia and other amino acids like leucine on NADPH?
⟶ The effects of hyperglycemia and amino acids on NADPH production are described in Section 3 of the manuscript.
- The translational aspect of targeting NADPH in T2DM is missing.
⟶ The translational changes at the protein level of enzymes that affect NADPH production are additionally described in Table 1.
- The effect of suppressing Nox2/NADPH on pancreas or the effect of knocking out Nox2 on insulin secretion and glucose levels should be discussed in a separate section.
⟶ I've added Section 8 and Table 3 to the document to better meet your needs.
- Islet cell transplantation has been suggested as a therapy to increase insulin production, what are the author's thoughts on this from the perspective of NADPH?
⟶ Transplanting islet cells into the body to help make more insulin is a good idea. It's worth thinking about how this affects NADPH, which is important for keeping cells safe and helping to make insulin. For these new cells to work right and make insulin well, they need to have enough NADPH, especially because diabetes can make cells more stressed. But remember, this is just what I think and I didn't include it in the paper.
- The role of Nox inhibitors should be discussed.
⟶ I've added Section 8 and Table 3 to the document to better meet your needs.
- There is no distinction between type I and type II DM. Like destruction of islet cells may be due to inflammation, a part has been discussed, but please relate it to NADPH. Further, inflammation also plays a role in TIIDM mainly in the presence of adipocyte hypertrophy-mediated inflammation. Thus, it is important to have two subsections, one for type I and one for type II.
⟶ Thanks for your helpful feedback. I've now included more information about both Type 1 and Type 2 diabetes in the paper. Just so you know, this paper mainly focuses on Type 2 diabetes.
- Including the translation aspect by discussing the role of and probability of targeting NADPH in atherosclerosis and other cardiovascular diseases and in renal disease will increase the readership and will make the review more interesting.
⟶ There's not a lot of research on how protein levels change. I've tried to include as many studies as we could in Table 1. Also, I've added information about NADPH's role in heart and kidney diseases in the conclusion part of the paper.

Reviewer 2 Report
Comments and Suggestions for Authors
The review article submitted by Moom D entitled "NADPH dynamics: Linking insulin resistance and beta cell ferroptosis in diabetes" is an excellent effort to discuss the importance of NADPH in light of the recent findings. My specific comments are as under:
1. In the title the author has used the term diabetes, which is misleading. The author should specify diabetes as diabetes mellitus.
2. In the abstract section Line: 8, the word comprehensive should be removed.
3. The introduction did not define or explain diabetes mellitus, instead, the introduction started with insulin resistance. The author should dedicate a few lines to describe diabetes mellitus for non-clinical readers.
4. The description of Figure 1. should be more comprehensive.
5. The bibliography used in the review is quite limited. The author should consider this because using a single study to justify a scientific claim is usually not the best way for scientific reviews.
Author Response
Reviewer 2
The review article submitted by Moom D entitled "NADPH dynamics: Linking insulin resistance and beta cell ferroptosis in diabetes" is an excellent effort to discuss the importance of NADPH in light of the recent findings. My specific comments are as under:
⟶ Thanks a lot for your detailed review. I tried my best to make changes in the short time I had. Just a heads up, I'll be on a business trip from December 22nd for three days where I can't use any electronics due to security reasons. If there's a need for more revisions, please keep my travel dates in mind.
- In the title the author has used the term diabetes, which is misleading. The author should specify diabetes as diabetes mellitus.
⟶ The title has been modified as below.
NADPH Dynamics: Linking Insulin Resistance and Beta Cell Ferroptosis in Diabetes
NADPH Dynamics: Linking Insulin Resistance and Beta Cell Ferroptosis in Diabetes Mellitus
- In the abstract section Line: 8, the word comprehensive should be removed.
⟶ I removed the word 'comprehensive' from the abstract.
This comprehensive review paper presents a detailed analysis
This review paper presents a detailed analysis
- The introduction did not define or explain diabetes mellitus, instead, the introduction started with insulin resistance. The author should dedicate a few lines to describe diabetes mellitus for non-clinical readers.
⟶ The following content has been added to the introduction.
“Diabetes mellitus is a widespread metabolic disorder where the body experiences elevated blood glucose levels due to either insufficient insulin production or the body's impaired ability to use insulin [1, 2]. Type 1 diabetes mellitus, an autoimmune condition, occurs when the body's immune system mistakenly attacks and destroys insulin-producing cells in the pancreas, leading to a critical deficit of insulin [3, 4]. In Type 2 diabetes mellitus, the body develops resistance to insulin, and despite having adequate or increased levels of this hormone, it becomes less effective at managing blood glucose levels [3, 4].”
- The description of Figure 1. should be more comprehensive.
⟶ The following has been added to Figure 1 Legend.
(A) depicts the ROS scavenging mechanisms where NADPH is utilized in the enzymatic reaction that regenerates reduced glutathione (GSH) from its oxidized form (GSSG) via glutathione reductase (GR). This reduced GSH then aids glutathione peroxidase (GPX) in detoxifying hydrogen peroxide (H2O2) into water, a vital process for controlling oxidative stress within the cell. (B) contrasts with the antioxidative role by illustrating NADPH as a substrate for the NADPH oxidase complex, which catalyzes the production of superoxide radicals (O2-). These radicals are key signaling molecules that activate various kinases such as JNK, P38, IKK, and PKCδ, thereby implicating NADPH in the modulation of cellular stress responses. (C) transitions into the realm of lipid metabolism, detailing NADPH's essential function in the synthesis of fatty acids. It traces the metabolic flow from glucose to pyruvate, then to acetyl-CoA, and its subsequent carboxylation to malonyl-CoA, followed by elongation to palmitoyl-CoA via fatty acid synthase (Fas), a process reliant on NADPH. The series of reactions leading to the formation of triacylglycerol accentuates the role of NADPH as a reducing agent in the biosynthesis of complex lipids. (D) outlines the cholesterol synthesis pathway, where NADPH again plays a crucial role as a reducing agent, particularly in the conversion of HMG-CoA to mevalonate by HMG-CoA reductase. The pathway is further detailed through subsequent conversions to intermediate compounds like isopentenyl pyrophosphate and farnesyl pyrophosphate, culminating in the production of cholesterol, underscoring the significance of NADPH in the synthesis of this vital lipid molecule.
- The bibliography used in the review is quite limited. The author should consider this because using a single study to justify a scientific claim is usually not the best way for scientific reviews.
⟶ Bibliography has been added for scientific reviews.